# Variability of Water Transit Time Distributions at the Strengbach Catchment (Vosges Mountains, France) Inferred Through Integrated Hydrological Modeling and Particle Tracking Algorithms

**Sylvain Weill \*, Nolwenn Lesparre, Benjamin Jeannot and Frederick Delay**

LHyGeS UMR 7517, Department of Earth Sciences, Université de Strasbourg, CNRS, ENGEES,
F-67000 Strasbourg, France; lesparre@unistra.fr (N.L.); bjeannot.pro@gmail.com (B.J.); fdelay@unistra.fr (F.D.)
\* Correspondence: s.weill@unistra.fr

**Abstract:** The temporal variability of transit-time distributions (TTDs) and residence-time distributions (RTDs) has received particular attention recently, but such variability has barely been studied using distributed hydrological modeling. In this study, a low-dimensional integrated hydrological model is run in combination with particle-tracking algorithms to investigate the temporal variability of TTDs, RTDs, and StorAge Selection (SAS) functions in the small, mountainous Strengbach watershed belonging to the French network of critical-zone observatories. The particle-tracking algorithms employed rely upon both forward and backward formulations that are specifically developed to handle time-variable velocity fields and evaluate TTDs and RTDs under transient hydrological conditions. The model is calibrated using both traditional streamflow measurements and magnetic resonance sounding (MRS)—which is sensitive to the subsurface water content—and then verified over a ten-year period. The results show that the mean transit time is rather short, at 150–200 days, and that the TTDs and RTDs are not greatly influenced by water storage within the catchment. This specific behavior is mainly explained by the small size of the catchment and its small storage capacity, a rapid flow mainly controlled by gravity along steep slopes, and climatic features that keep the contributive zone around the stream wet all year long.

**Keywords:** integrated hydrological modeling; mountainous catchment; particle tracking; storage selection functions; transit and residence time distributions

## 1. Introduction

Mountainous headwater catchments are widely used for water supply and bring important services to the downstream part of rivers through the export of water and associated biogeochemical and sedimentary fluxes (e.g., [1,2]). Understanding streamflow generation processes and delineating the main flow pathways in mountainous catchments are thus critical for addressing issues related to sustainable water resources and ecosystem management (e.g., [3,4]). Streamflow-generation processes in headwater catchments are complex because of the strong controls exerted by spatially variable topographic, geologic, pedologic, geomorphologic, and climatic features (e.g., [5,6]), and because of the transient interactions between the diverse compartments of the catchment. Mountainous subsurface structures are in general characterized by shallow soils on hillslopes that are connected to riparian areas close to streams [7]. The geometry and the variability of the subsurface structures are often poorly described, even though these characteristics strongly impact the evolution over time and space of the connectivity between the riparian areas and the hillslopes. This connectivity (which also partly determines the water storage within an active riparian zone) has been proven a key parameter for both

streamflow and water-quality responses (e.g., [8,9]). Connectivity is also one of the explanations for highly nonlinear storage–discharge relationships in mountainous catchments (e.g., [10,11]).

Despite the improvement of theoretical, experimental, and modeling approaches to surface and subsurface hydrology, our current understanding of streamflow-generation processes in mountainous headwater catchments remains limited (e.g., [12]). Further studying transit-time and residence-time distributions (hereafter denoted as TTDs and RTDs) could be a way to unravel some fundamental questions related to the hydrological response of catchments. TTDs and RTDs are considered to be key descriptors of how a catchment stores and releases water (e.g., [13,14]). They are also viewed as powerful tools for developing new theoretical approaches to catchment responses [12], and for bridging the gap between catchment hydrology and water-quality preoccupations (e.g., [15]). TTDs describe the distribution of the times spent by water parcels between their entry as rainfall (or any other process at the boundary conditions of the system) and their exit at a given time, mainly in shallow watersheds as streamflow or evapotranspiration. RTDs describe the distribution of the ages of water parcels that are stored (i.e., remain in the system) within a catchment at a given time. Both distributions can be estimated using either a forward or a backward formulation for the delineation of streamlines riddling the watershed and the calculation of times at which water parcels pass through a location along these streamlines [16]. Even though forward and backward approaches are expected to render different results or interpretations, few studies at the catchment scale have investigated the differences between these two kinds of formulations. Theoretical concepts have recently been proposed to account for the variation over time of TTDs [17–21]. The most widely used concept is based on the StorAge Selection (SAS) function [20], a single function that combines TTDs and RTDs to indicate how a catchment stores or releases water of different ages at a given time. This new indicator has been applied to various hydrological systems (e.g., [22–25]) and seems to be a promising approach to the consequences of flow and transport processes at the scale of a whole watershed.

Up to now, studies about the temporal variability of TTDs and RTDs have mainly relied on concentration measurements of conservative geochemical or isotopic tracers [14], the underlying idea being to employ conceptual models or convolution kernels to determine the shape of the distributions that properly relate input and output geochemical signatures (e.g., [18,22,26]). Studies of distributions based on catchment flow and transport modeling are also available, but these have only received consideration as valuable alternatives to tracer-based studies [27–34]. The hydrological models used in these studies range from simply-conceived rainfall–runoff relationships (e.g., [30]) to more complex, fully distributed models (e.g., [31–34]). Transport processes are described using either conceptual approaches or explicit calculations such as the Eulerian resolution of an advection–dispersion equation or Lagrangian particle-tracking simulations.

Physically based distributed-flow models combined with transport algorithms are appealing as they can simulate the response of catchments under various configurations—for example, those involving transient forcing terms and boundary conditions—and can be used to evaluate directly the evolution over time of TTDs and RTDs (e.g., [29]). Their application is more expensive in terms of computational costs, but they also render time-varying TTDs and RTDs in relation to the succession of physical processes and flow pathways that occur in the watershed. To date, few studies based on physically based distributed models have investigated the temporal variability of TTDs or RTDs on actual catchments using transient-flow conditions [31–34]. Most of the previous work on TTDs and RTDs based on catchment modeling have considered saturated-groundwater flow alone (e.g., [35]), and/or steady-state flow conditions. Integrated surface-subsurface hydrological models (i.e., models that explicitly couple surface and subsurface processes) are ideal tools to address the issue of time-varying RTDs and TTDs, but so far, they have only been applied to synthetic test cases [29,33] to assess the potential effect of subsurface heterogeneity, land cover, and climate.

In this study, a low-dimensional integrated hydrological model, named the Normally Integrated Hydrological Model (NIHM) [36–39], is employed to simulate the hydrological response of a mountainous headwater catchment (the Strengbach catchment, in the Vosges Mountains) belonging to

the French network of critical-zone observatories. The low-dimensional approach developed in NIHM enables researchers to describe coupled surface-subsurface flow in a physically consistent manner while reducing significantly the computational cost involved. This approach is especially relevant when dealing with natural systems that would require a very dense computational mesh with classical 3D approaches—i.e., systems with steep topography that require a refined vertical discretization to properly describe vertical water flow or systems with a flat topography that require a refined horizontal resolution to describe complex small-scale topographic features. Particle-tracking algorithms are used to simulate advective transport within very transient flow fields and explicitly evaluate the temporal variability of TTDs, RTDs, and SAS functions. Both forward and backward formulations are implemented with the aim of comparing the effects of the formulations on the calculated distributions of times. With this approach, we are interested in (i) providing new insights into the evolution over time of TTDs and RTDs and thus improving our overall understanding of headwater catchment functioning; (ii) investigating the relationship between this evolution and the amount of water stored in the catchment; and (iii) addressing questions related to the differences between time distributions calculated via forward and backward particle-tracking formulations.

The paper is organized as follows. Section 2 summarizes the presentation of the studied area and its main hydrological characteristics. Section 3 is dedicated to the description of the methods employed—namely, the hydrological model NIHM, and the particle-tracking algorithms for transient flow fields. Section 3 also details the methodology used to calibrate and verify the model against observed data and how to evaluate the temporal variability of TTDs, RTDs, and SAS functions. The results obtained for the Strengbach catchment are presented and discussed in Section 4.

## 2. Materials and Methods

### 2.1. Study Site

The Strengbach catchment is a small, forested headwater catchment of 0.8 km$^2$ surface area located on the upper crests of the Vosges mountains, in Northeast France [40]. The elevation of the area ranges between 880 and 1150 m with heavily incised side slopes, especially in the part of the catchment close to the permanent stream in the bottom of the valley (Figure 1). The local climate is of the temperate oceanic type with an average rainfall during the period 1986–2015 of 1380 mm/year, inter-annual variations from 890 to 1710 mm/year, and a mean potential evapotranspiration of 570 mm/year associated with inter-annual variations from 510 to 730 mm/year. Mean monthly precipitation over the Strengbach catchment is approximately 115 mm with weak variations over the course of the year. The mean annual discharge at the outlet over the same period was around 750 mm/year with inter-annual variations ranging from 490 to 1130 mm/year. The mean discharge at the outlet was 20 L/s with an instantaneous discharge that ranged from 1 to 400 L/s.

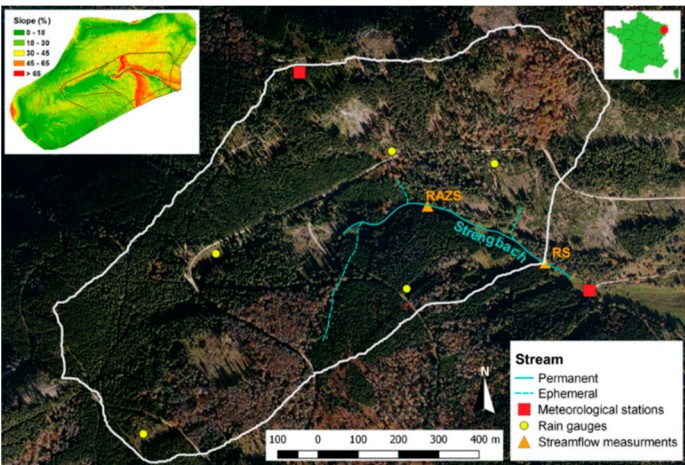

**Figure 1.** Map of the Strengbach catchment with the stream network and the equipment used for meteorological, geochemical, and hydrological monitoring. The land slope (%), derived from a high-resolution LIDAR survey (0.5 × 0.5 m), is shown in the upper left frame.

The bedrock of the catchment is mainly granitic with small outcrops of gneiss along the northern boundary. Weathering generated a strong alteration of the bedrock and the development at the land surface of a saprolite of 1 to 9 m thickness (the average thickness is 4 m). The soil covering the catchment is coarsely grained and sandy with relatively high saturated hydraulic conductivity (on the order of $10^{-4}$ m·s$^{-1}$). Ninety percent of the surface area of the catchment is covered by a forest with two main species, spruce and beech shafts; these two species represent 80% and 20% of the total tree population, respectively.

Hydrological, geochemical, and climatic parameters have been monitored at the catchment since 1986 at various scales and with a dense network of sensors (see [40] for details). The location of the equipment necessary to collect the data employed in this study is specified in Figure 1. The meteorological stations and rain gauges are used to estimate rainfall and evapotranspiration. The average rainfall over the catchment is calculated by weighting the rainfall measure at the upper meteorological station with a correction coefficient. This coefficient is set up by comparing the rainfall value at the upper station with the spatialized rainfall interpolated from the rain-gauge measurements. Evapotranspiration is estimated via the Penman empirical formulation and standard climatic measurements (wind speed, air moisture level, solar radiation, and so forth). The two streamflow measurements (RAZS, 400 m upstream from the outlet, and the outlet RS, as shown in Figure 1) are used for the purpose of model calibration and/or verification.

Magnetic resonance soundings (MRS) were also carried out to identify the evolution of the subsurface water storage at various locations within the catchment. The principle of MRS is to activate and then cut off an alternating current in a wire loop on the ground surface, thereby generating an MRS signal for water molecules (e.g., [41]). The same loop is then used to measure the signal emitted by the energized water molecules beneath the loop. Several activations of different intensities are repeated for a given location with the aim of exploring the subsurface at different depths. The inversion of the MRS signal allows for estimates of the saprolite thickness, the quantity of water beneath the measurement points, and the permeability of the porous medium (e.g., [42]). MRS measurements were performed once at each station between April and May 2013 [40]. Data were collected using eight-shaped loops 37.5 m in length. The initial distribution of the measurement stations was defined to cover the catchment uniformly. However, because an ambient magnetic noise may have affected the quality of the data, some measurement points, especially for those located on the northern slope, were not considered in this study. The MRS signal depends on the underground water content distribution at various depths and was directly used as a basis for the NIHM calibration.

Previous studies at the Strengbach catchment showed that the hydrological response of the system at its outlet was strongly controlled by the dynamics of the saturated area located close to the stream (e.g., [43]). Although this saturated area represents only a small fraction of the total area of the catchment (3%), the dynamics of the water table close to the stream produce most of the streamflow, with a high contribution of pre-event water to the total instantaneous outflow [44]. Further details on previous studies carried out at the Strengbach catchment are summarized in [40].

*2.2. The Normally Integrated Hydrological Model (NIHM)*

The model NIHM used to simulate the response of the Strengbach catchment is a physically based spatially distributed model that describes and couples the flow processes occurring in the surface and subsurface compartments of a watershed [36–39]. The model can describe flow in the subsurface, in a 1-D stream network, and also over the 2-D land surface. As diffuse 2-D surface run-off or exfiltration from the subsurface have never been observed at the Strengbach catchment, only 1-D routing via the stream network and subsurface flow are presented below.

The originality of NIHM lies in the way it describes the subsurface flow. The equation used for subsurface flow results from the integration of the full 3-D Richards equation along a direction perpendicular to the bedrock. Flow within the 3-D subsurface domain is thus modeled through a 2-D equation and mean parameters as local integrals over the saturated and unsaturated thicknesses of the subsurface. This technique drastically reduces the meshing effort and the computational cost while preserving the physics, especially for systems with contrastive topography, as is the case with the Strengbach catchment discussed here. Conversely, the technique also works for flat systems needing a refined mesh to capture small topographic depressions, as exemplified by a riverine island [38,39]. The description of flow in the 1-D stream network is based on a 1-D formulation of the diffusive wave approximation of the St-Venant equations. The set of equations that are solved can be written as:

$$\frac{\partial \overline{\theta}}{\partial t} + \overline{S}(h)\frac{\partial h}{\partial t} + \nabla_{x,\,y}\left(-\overline{\mathbf{T}}(\theta)\nabla_{x,\,y}h\right) = Q_w \tag{1}$$

$$l_r\frac{\partial h_r}{\partial t} - \nabla_x(k_p\nabla_x(h_r + z_r)) = l_r\gamma_{1D} \tag{2}$$

With

$$\overline{\theta}(h) = \int_{z_w}^{z_s} \theta(z)dz \; ; \; \overline{S}(h) = S_{sat}\,h \tag{3}$$

And

$$\overline{\mathbf{T}}(h,\theta) = \mathbf{K_{sat}}\,h + \int_{z_w}^{z_s} \mathbf{K}(\theta(z))dz \tag{4}$$

$\mathbf{K_{sat}}$ and $S_{sat}$ are averages along the integration direction $z$ of the saturated hydraulic conductivity tensor and the specific storage capacity in the saturated zone, respectively. $\theta$ [-] is the water content; K [LT$^{-1}$] is the tensor of hydraulic conductivity; $h$ [L] is the hydraulic head (or the capillary head) in the subsurface; $Q_w$ [LT$^{-1}$] is a sink/source term that accounts for the subsurface interactions with the 1-D river network; and $z_w$, $z_s$ [L] are the local coordinates along the direction $z$ normal to the bedrock of the water table and land surface, respectively. $h_r$ [L] is the water depth in the river network; $z_r$ [L] is the elevation of the river bed; $l_r$ is the wetted width in the river; $k_p$ [L$^3$T$^{-1}$] is a conduction term that depends on geometrical parameters of the network, the Manning roughness coefficient, and the surface water depth $h_r$; and $\gamma_{1D}$ is a sink/source term accounting for the stream network interactions with the subsurface.

The coupling between the surface and subsurface is performed using a first-order approximation that sets the exchanged fluxes between compartments as proportional to the head difference between the compartments. The equations are solved together using a fully implicit approach and advanced numerical schemes. A full description of the model and the numerical techniques used for the resolution

is available in [38]. The coupled model was carefully verified using model inter-comparisons and bench tests [37,38], and was proven relevant when applied to actual complex hydrosystems [39].

The indicators used to assess the efficiency of the model when applied to the Strengbach catchment are the Nash–Sutcliffe efficiency (NSE) coefficient, the Kling–Gupta efficiency (KGE), and the root mean square error (RMSE) are defined as follows:

$$NSE = 1 - \frac{\sum\limits_{t=1}^{n} (Q_{s,t} - Q_{o,t})^2}{\sum\limits_{t=1}^{n} (Q_{o,t} - \overline{Q}_o)^2} \tag{5}$$

$$KGE = 1 - ED; \ ED = \sqrt{(r-1)^2 + (\alpha-1)^2 + (\beta-1)^2} \tag{6}$$

$$RMSE = \sqrt{\frac{\sum\limits_{t=1}^{n} (Q_{s,t} - Q_{o,t})^2}{n}} \tag{7}$$

where $n$ is the number of observation times; $Q_{s,t}$ is the simulated discharge at time t; $Q_{m,t}$ is the observed discharge at time $t$; $\overline{Q}_o$ is the mean of the observed discharge; $\alpha$ is the ratio between the mean simulated and mean observed discharges; $r$ is the correlation coefficient between the simulated and observed discharges; and $\beta = \frac{\sigma_s}{\sigma_o}$ is the ratio between the standard deviations of the simulated and observed discharges.

### 2.3. Particle Tracking Algorithms

A particle-tracking model based on a classical Lagrangian approach was developed to derive TTDs and RTDs. Particles are launched into the computational domain and moved by advection only through the transient velocity fields that are simulated by NIHM. Although NIHM works with finer time steps, the velocity fields and the length of the active-stream network are exported only with a daily time interval for convenience and a reduction of calculation costs when tracking the particles. The evolution of the location $X_k$ along a one-dimensional curvilinear streamline for a given particle $k$ is computed by:

$$X_{k,next} = X_{k,start} + V\big(X_{k,start}\big)\Delta t_k \tag{8}$$

where $X_{k,next}$ is the location at the next time step, $X_{k,start}$ is the starting location, $V(X_{k,start})$ is the velocity vector in the cell of the current location of particle $k$, and $\Delta t_k$ is the time taken by the particle to move from $X_{k,start}$ to $X_{k,next}$. In practice and as illustrated in Figure 2, a particle located in a current cell moves according to the velocity vector of the cell until one edge of the cell is reached. When on the edge, (1) the location $X_{k,next}$ and arrival time are recorded; (2) the location $X_{k,start}$ is updated; (3) the velocity vector of the particle is also updated to reflect the value and direction of the neighboring cell; and (4) the particle displacement is resumed. The time step $\Delta t_k$ varies over time because of the space and time variability of the velocity field and also because of the geometrical characteristics of the computational mesh. Therefore, $\Delta t_k$ is computed independently for each particle and each jump depending on its flow path. It is worth noting that for transient-flow conditions, the whole velocity field is also updated (here with a daily time interval), making it possible for a particle to be located within a cell (and not at an edge) as the velocity field changes. All the times and positions when a particle hits an edge or when the velocity field is updated are stored, allowing the positions of all the particles to be determined at a given time.

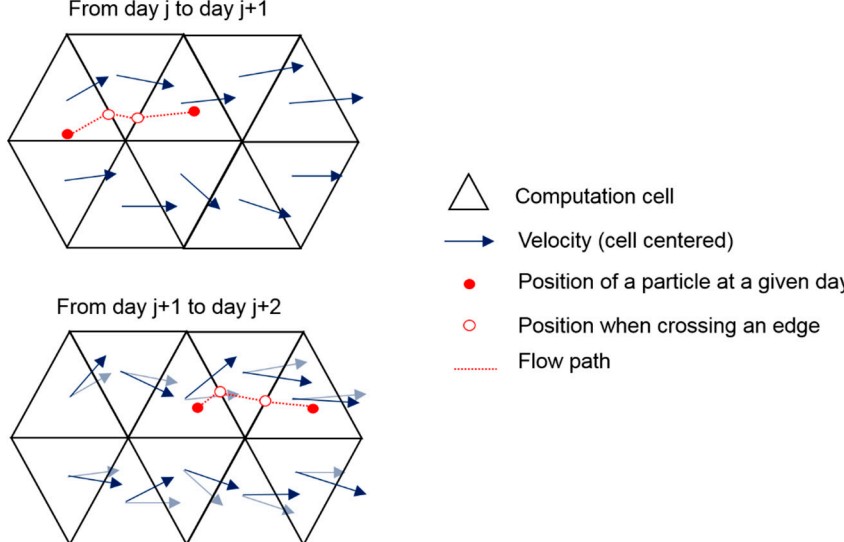

**Figure 2.** Illustration of forward particle-tracking procedure for a single particle moved within transient velocity fields with daily update.

Both forward and backward formulations are implemented in the model. While RTDs are estimated only via the forward formulation, TTDs can be derived using both formulations. Figure 3 sketches out the difference between forward and backward formulations. In the forward form, the particles are injected across the land surface. Each particle then moves within the domain following the daily-updated velocity field simulated by NIHM until it reaches the active-stream network considered as the exit for the particle. This approximation does not bias the RTD and TTD calculations, because the transit times in the small-stream network of the Strengbach are much shorter than in the subsurface (transit times in the stream are on the order of several hundred seconds). In the forward approach, when it is not the aim to draw the streamlines along which the particles are moved, the injection and exit time of each particle injected are sufficient to estimate the TTDs and RTDs and their variability over time (further details are provided below). In the backward formulation, particles are injected close to the active-stream network at a given time. Each particle then moves into the domain following a reverted velocity field—i.e., a field that has the same amplitude as the one simulated by NIHM but that moves in the opposite direction—until it reaches the boundaries (or eventual dead ends) of the domain. As for the forward approach, the velocity fields are updated every day to account for transient flow in the system. In the backward approach, the positions of the particles at all times between injection and exit times must be recorded to compute the backward TTDs. If only injection and exit times were kept, the TTDs would be biased, because they would correspond to transit times conditioned by the fact that water only entered the system through its lateral boundaries (or dead ends).

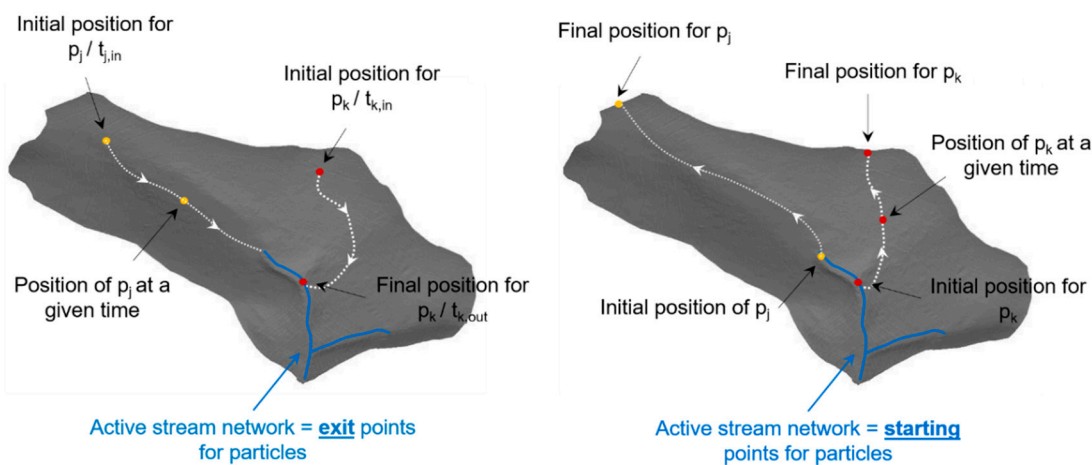

**Figure 3.** Illustration of forward and backward particle-tracking formulations. Only two particles are represented to avoid overly detailed sketches.

### 2.4. Model Setup, Calibration and Verification

The computational mesh used both for the simulations with NIHM and for the particle-tracking calculation is generated from a fine LIDAR survey of $0.5 \times 0.5$ m resolution. The resulting computational domain of 10,250 elements is shown in Figure 4. The resolution is much finer close to the stream and close to the locations where MRS measurements are available, with an elementary cell size of 1 m. This refinement helps properly capture the hydrological processes in the hyporheic zone close to the stream, and to obtain a high resolution over space of the subsurface storage close to MRS measurements. On hillslopes, where flow is mainly triggered by gravity forces (and rainfall inlets), a cell size of approximately 20 m is enough for an accurate depiction of the flow field.

The main parameters to set up in NIHM for the specific simulation of the Strengbach catchment are those of the subsurface. It is worth restating that no 2-D diffuse surface flow occurs, and also that, given the steep slopes of the 1-D stream network, surface routing is sensitive to its geometrical settings (well-defined by the LIDAR image) but not its flow parameters. The subsurface must be defined with respect to: the depth of the different horizons, their respective porosity, the saturated hydraulic conductivity, and the Van Genuchten parameters for the relationships between water content, effective hydraulic conductivity, and capillary pressure in the vadose zone. In this study, the Van Genuchten parameters $\alpha$ and $n$, along with the saturated hydraulic conductivity, are assumed to be uniform over the whole domain, with values of $2$ m$^{-1}$ and 1 for $\alpha$ and $n$, respectively, and $10^{-4}$ m/s for the saturated hydraulic conductivity. Only the porosity and the thickness of the different subsurface layers are considered as spatially variable. The subsurface domain is composed of two layers; the upper one is heavily weathered and thin, while the deeper one is less porous and thicker. The spatial distribution of the subsurface layers is based on a field study dedicated to mapping the different types of soil present in the catchment. Some information about the porous-formation properties is available from the analysis of core samples at a few locations in the catchment. Figure 4 shows the 8 zones considered as sub-areas of the uniform subsurface hydraulic properties that cover the whole catchment. The delineation of these sub-areas was not modified by the calibration procedure. The values of the porosity and thickness of the two layers were first set using intuitive but reasonable initial values. Then, they were calibrated "manually" following a simple Monte Carlo procedure, which involved sampling multiple parameter sets around an initial guess and then selecting solutions that best fit the available data on streamflow at the outlet of the catchment and the local MRS measurements sensitive to water content in the subsurface. The final calibrated values of porosities and thicknesses of subsurface layers in each zone of the domain are reported in Table 1.

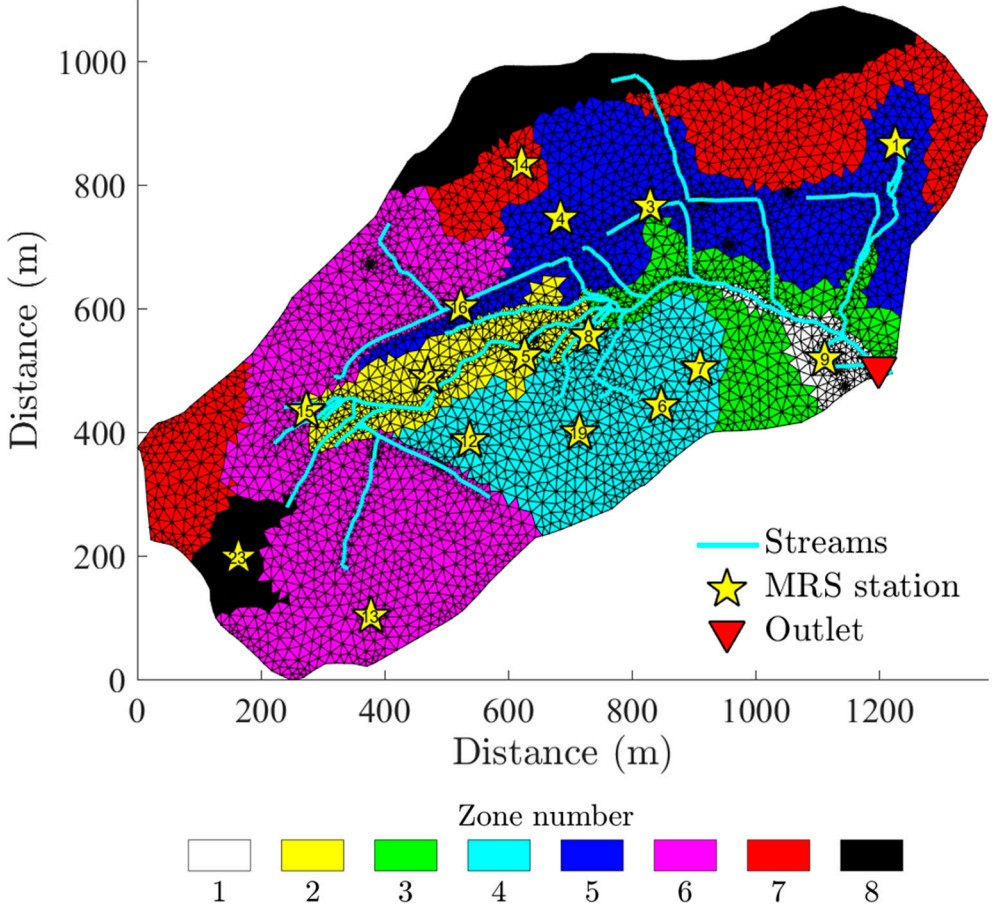

**Figure 4.** Delineated zones used for the spatial distribution of the subsurface hydrodynamic parameters (North direction along the vertical axis). Yellow stars represent the magnetic resonance sounding (MRS) measurement points. The computational mesh is also represented, illustrating the refinement close to the stream and the MRS measurement points.

**Table 1.** Calibrated values of thickness and porosity in the two layers used to describe the subsurface domain for each zone (as presented in Figure 4).

| Zone Number | Total Thickness [m] | Thickness Layer 1 [m] | Thickness Layer 2 [m] | Porosity Layer 1 [−] | Porosity Layer 2 [−] |
|---|---|---|---|---|---|
| 1 | 10 | 2 | 8 | 0.5 | 0.03 |
| 2 | 14 | 1 | 13 | 0.5 | 0.03 |
| 3 | 4 | 1.5 | 2.5 | 0.6 | 0.02 |
| 4 | 3 | 1 | 2 | 0.3 | 0.02 |
| 5 | 7 | 1 | 6 | 0.1 | 0.02 |
| 6 | 4 | 0.5 | 3.5 | 0.1 | 0.02 |
| 7 | 2 | 1 | 1 | 0.5 | 0.02 |
| 8 | 3 | 0.5 | 2.5 | 0.2 | 0.05 |

The lateral boundaries of the domain are considered as no-flow boundaries; water can only flow out of the system at the stream mouth down in the valley of the catchment. An effective rainfall history, with values consisting of raw rainfall totals minus potential evapotranspiration estimated using the empirical Penman equation, is applied on the surface of the catchment to describe the evolution of the meteorological forcing. The initial conditions for the calibration and verification simulations were set by letting a simple drainage procedure occur over a system with initially prescribed hydraulic heads. As the topography is very steep and the catchment very reactive, this approach is not sensitive to the prescribed water-saturated thicknesses and furnishes data sufficient to provide a reasonable initial

state. In addition, the comparison of model outputs with measurements is performed for periods starting a few months after the beginning of the simulations, so that the eventual bias due to initial conditions is dampened by the simulation time and does not unduly affect evaluation of the solutions.

The calibration of NIHM for the Strengbach catchment is not performed in a traditional way, i.e., by relying solely on hydrological variables such as streamflow to compare model outputs and observations. Instead, hydrogeophysical information from MRS measurement is also considered, to better constrain the set of parameters and reduce equifinality in the model outputs—a problem that can occur due to a lack of conditioning information when only streamflow is considered. The MRS measurements exploited at the Strengbach catchment provide the evolution over short times (here, on the order of 150 milliseconds) of the electric potential (in millivolts) associated with the magnetic field in response to several current intensities activated in the subsurface. An exponential decrease is traditionally used to describe this evolution. The objective of the calibration procedure is to fit the model both to the streamflow and to the MRS measurements.

Although the hydrological modeling with NIHM renders streamflow evaluations at the outlet, it does not provide a direct evaluation of MRS measurements. Nevertheless, NIHM provides the local distribution over depth of the subsurface water content, which is the sensitive variable for MRS measurements. MRS "indirect" simulations stemming from the hydrological modeling are therefore direct calculations of the MRS magnetic signal associated with water contents simulated by NIHM. In this study, the goodness of fit for MRS is evaluated through a comparison between the integrals of the measured and simulated signals for the different current intensities. Stated slightly differently, maps of water content simulated by NIHM at the date of MRS acquisitions are post-processed to estimate the MRS signal, which is then compared to the MRS measures. The set of parameters in NIHM that renders the best fit of MRS measurements and the best traditional indicators on streamflow (here, the KGE coefficients) is selected for the step of model verification and then for further particle-tracking simulations.

The period used for the calibration step runs from 1 October 2012 to 31 May 2013. It includes a 4-month "warm up" of the model (see our previous discussion of the point related to initial conditions) and then 4 months during which observed and simulated streamflow histories at the outlet are compared. Once the best set of parameters has been identified, two simulations (verifications) are carried out, allowing for a check of how the model behaves over hydrological periods that did not serve as reference periods for the model calibration. The first verification covers the period between the 1 March 1996 and 31 July 1996. It was chosen because it is the only short period during which measurements of discharge in the stream are available both at the outlet of the catchment and 400 m upstream (the RAZS location in Figure 1). A second verification simulates the system over a nine-year period (2008 to 2016), with a view to evaluating the ability of the model to reproduce the long-term response of the catchment. For this period, only measurements of the discharge at the outlet are available; these measurements are used to compute the KGE coefficients and RMSE. The various particle-tracking simulations are performed only within this longer-term period.

It is worth noting that the Monte Carlo procedure employed to test various configurations of the catchment does not ensure that the selected configuration is optimal, in the sense that both the parameterization and the parameter values render the best fit between model outputs and data. It is likely, however, that an automatic inversion procedure, relying upon optimization to minimize the residuals between simulations and data, would not perform better. At the Strengbach catchment (as with many other under-sampled actual systems), the main measure characterizing the hydrological behavior of the catchment is the discharge of the stream at the outlet. This measure obviously aggregates (convolutes) the responses to the local hydraulic behaviors of the system. Calibrating the model on this response results in multiple configurations able to mimic observations. That being said, the selected configuration reproduces fairly well the overall behavior of the Strengbach catchment in terms of streamflow at the outlet (see Section 4). As the TTDs and RTDs are also indicators of the overall

behavior of a hydrological system (as opposed to indicators of local functioning), one can be confident that the model renders reliable information about the various times spent by water in the catchment.

*2.5. Estimation of RTDs, TTDs, and SAS Functions*

Estimating the temporal variability of forward RTDs and TTDs requires that particles be injected repeatedly, so that all the possible ages of particles stored in the catchment can be sampled properly. Water parcels do not enter the system at the same date or the same location. The choice was made to inject one particle per computational cell (there are 10,250 cells in the computational domain) each time the daily cumulated effective rainfall was greater than 5 mm, resulting in 670 particle injections during the 2008–2016 period. The particle also holds a weight proportional to the intensity of effective rainfall, so that times associated with the particles will have a probability density proportional to the mass of water injected. Overall, approximately 6.8 million particles were launched during this period. All these particles were tracked using the forward algorithm described previously. The model provides for each particle $k$ its injection time $t_{k,in}$ and its exit time $t_{k,out}$. The RTDs and TTDs can then be computed at a given date of interest $t_{obs}$ using the following approach:

- A first selection is made regarding the value of the injection time $t_{k,in}$ for each particle. If $t_{k,in} > t_{obs}$, the particle can neither be stored in the catchment nor reach the active-stream network at the given date; thus the particle should not be considered for the RTDS and TTDS estimations. Only the particles with an injection time earlier than $t_{obs}$ are selected.

- A second selection is made regarding the exit time $t_{k,out}$ for the particles that passed through the first selection. If $t_{k,out} < t_{obs}$ the particle has already left the system; it is no longer stored in the catchment and should not be considered for the RTDs estimation. Only the particles with an exit time $t_{k,out} > t_{obs}$ are considered when estimating the RTDs with a residence time equal to $t_{k,in} - t_{obs}$. If $t_{k,out} \sim t_{obs}$ (up to a small tolerance limit for $t_{obs}$), the particle has reached the active-stream network at the date of interest and should be considered in the TTDs estimate with a transit time equal to $t_{k,out} - t_{k,in}$.

This approach produces two classes of particles that are then used to compute for each date of interest $t_{obs}$ standard and cumulative RTDs and TTDs. As previously mentioned, the weight given to each particle—a weight that is proportional to the effective rainfall intensity—is used to weight the times of a particle in the calculation of the distributions, the probability of a given time simply being the ratio of its weight to the sum of weights collected in the distribution. The SAS function is computed and represented using the cumulative RTDs and TTDs.

Backward TTDs are also estimated using the backward particle-tracking algorithm described previously. At a given date of interest, the particles are launched close to the active-stream network. Much attention was given to the initial locations of the particles, because they can strongly impact the estimated TTDs. Particles were placed on both sides of the active-stream network to ensure a proper sampling of the catchment. Lack of information about the diverse flow-rate values along the stream did not allow us to weight the injected particles. Thus, all the particles convey the same mass of water, which could partly bias the TTDs. The backward formulation has the advantage of injecting fewer particles. It delineates easily the longest streamlines cutting through the system (particles stopped at dead-ends or at the boundaries of the domain), and defines "exact" times of interest for calculating distributions. By contrast, a forward approach is more tolerant about time of interest, because it needs to sample enough particles (see above). However, the backward approach does not discriminate the successive locations of the particles between an inlet in the system and a transit point along a flow line. When evaluating backward TTDs on the basis of the locations and associated times of the particles, each point and each time must be considered as the injection of the particle. This assumption is obviously wrong, but it accords with considering the whole catchment as being uniformly watered by rainfall over time and space.

## 3. Results

### 3.1. Model Performance

Figure 5 displays the evolution over time of the simulated and observed discharges at the stream outlet for the calibration period (a), and the comparison between the simulated and measured integrals of the MRS signals for different current intensities at the two stations #5 and #23 (see the panel (b) of Figure 5; the location where the measurements were taken is shown in Figure 4). The evolution of the discharge at the outlet shows that the model reproduces the overall response of the system with good accuracy. The efficiency coefficients quantifying the fit between model and data are high, with a KGE coefficient of 0.91, NSE coefficient of 0.91, and RMSE of 6.36 L/s. The simulated and measured integrals of MRS signals at the selected stations are also in good agreement, as are the responses at other stations (data from the other stations are not reported here). This feature shows that the calibrated model is also able to properly describe the local evolution of water contents over depth within the subsurface. It is worth noting that preliminary calculations using a fully uniform parametrization over the whole catchment for all the hydrodynamic parameters also allowed us to simulate with a good accuracy the evolution of the streamflow at the outlet and at RAZS (when available). This parametrization was then evaluated using the MRS measurements: It failed to reproduce the measured MRS signals. This result suggests that there are equifinality issues in the model parameterization when one relies only on discharge measurements to condition the model. MRS measurements provide valuable additional constraints that illuminate the local behavior of the system and thus result in a model with higher reliability. Employing MRS measurements in the calibration procedure is a true added-value, because it partly reveals the heterogeneity of the catchment and its local responses, whereas the streamflow measurement only depicts an integrated response of the overall dynamics of the catchment.

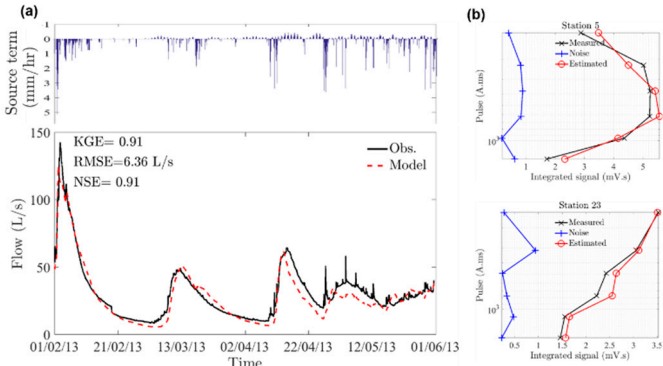

**Figure 5.** Evolution over time of simulated and observed streamflows for the calibration period (**a**) and comparison between simulated and measured integrals of the MRS signals at stations 5 and 23 (**b**).

Figure 6 shows the comparison of simulated and measured discharges at the outlet and at the RAZS station for the two verification steps. (To reiterate, the RAZS station was availably only during the first verification period.) Note that there are missing values in the RAZS streamflow chronicle associated to the breakdown of the measurement device. Even though the efficiency coefficients are smaller than for the calibration period, these coefficients (KGE, NSE, and RMSE) are still high. Although the matching of model outputs and data is good, it is noticeable that for both verification periods, the model tends to slightly overestimate the peaks of streamflow and underestimate the discharge during recession periods. This model behavior was foreseeable as a consequence of the main assumptions about flow that allowed us to build the low-dimensional subsurface compartment in NIHM [36]. Even though the amplitude of the peaks is not always perfectly reproduced, no significant time lags between the simulated and observed hydrographs occur, showing that the overall diffusion times for flow at the scale of the catchment are well-reproduced. The slight discrepancies between simulated and observed discharges at RAZS can be explained by the specific location of the measurement point, just downslope

from a small, localized contributing saturated area. Capturing the flat head gradient in such zones surrounded by steep topographic slopes is difficult. Small discrepancies could easily result in either the underestimation of discharge during recessions, because of head gradients that are slightly too flat, or the overestimation of discharge during periods of peak flow, as a consequence of a sudden increase in the simulated head gradients. Finally, the comparison of simulated and observed discharges for the verification periods remains convincing, resulting in a model able to capture the flow dynamics of the entire catchment and to render valuable estimates of transit-time and residence-time distributions.

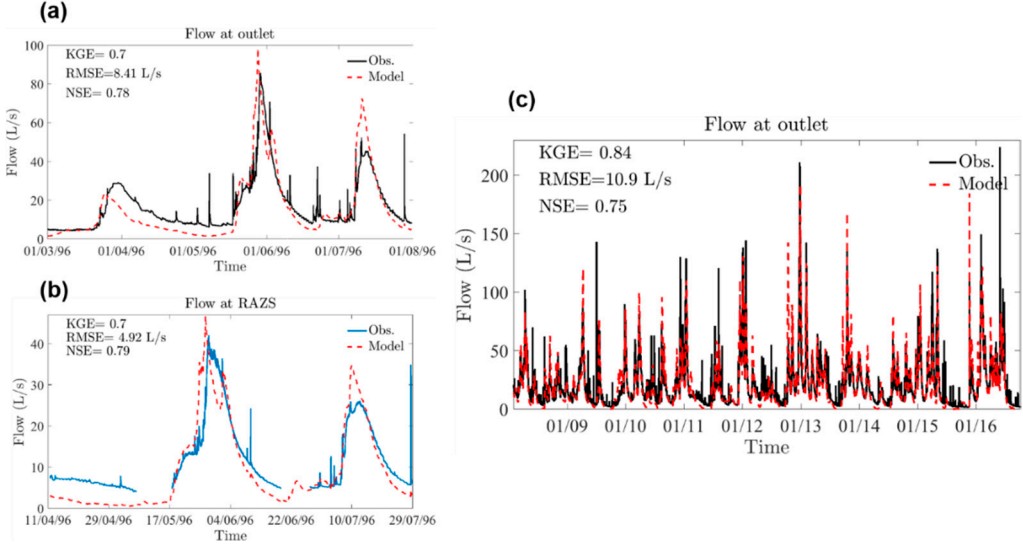

**Figure 6.** Evolution over time of simulated and observed streamflows. For the first verification period, streamflows at the outlet are shown in (**a**) and at those at the RAZS location are shown in (**b**). For the second verification period, streamflows at the outlet are shown in (**c**).

The evolution over time of water storage in the catchment is plotted in Figure 7. For better readability, the plot reports on a relative variation (in mm of water) with 1 January 2008 arbitrarily selected as a reference 0. This storage evolution was used to fix, respectively, 15 and 8 reference dates to estimate TTDs and RTDs under both high-storage (HS) and low-storage (LS) conditions. The relative storage does not show any regular trend from one year to another, with HS and LS conditions appearing at different periods in a given year. The two examples of years 2009 and 2013 (Figure 7) indicate high-storage periods in April and September 2009 versus February and October 2013, and low storage conditions in July and October 2009 versus August 2013. That being said, the water storage remains stationary over time for the 9 years of the studied period. The relative storage is of small variability over time, oscillating between +100 and −60 mm, but it is worth noting that there was no severe drought during this period. This small variability can be explained by the specific climate conditions encountered in this part of the Vosges Mountains, where a cumulative rainfall of approximately 120 mm falls almost every month. The catchment is thus constantly under relatively wet conditions, and, as discussed in what follows, this peculiarity strongly impacts the temporal variability of TTDs and RTDs.

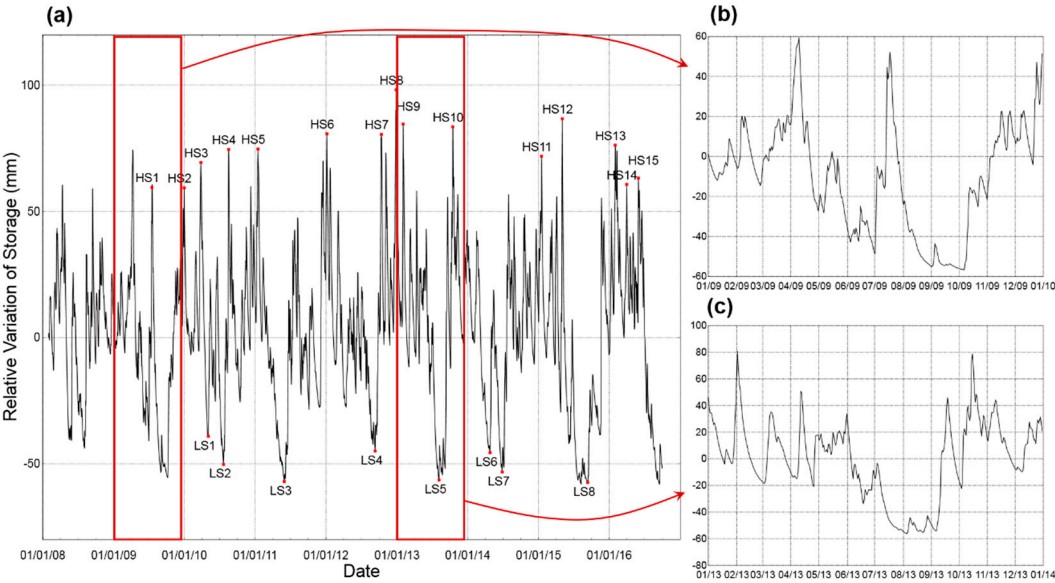

**Figure 7.** Evolution of water storage within the whole catchment during the period 2008–2016 (**a**). The labeled peaks in the plot correspond to reference dates at which residence- and transit-time distributions were estimated. HS stands for high-storage periods and LS for low-storage periods. Focused plots illustrate the seasonal and inter-annual storage variability for the years 2009 (**b**) and 2013 (**c**).

### 3.2. TTDs, RTDs, and SAS Functions and Their Relationship to Storage

Maps in Figure 8 illustrate the flow paths and transit times estimated with the backward particle-tracking algorithm at two different dates, i.e., the high-storage date HS9 and the low-storage date LS6. The delineation of the main flow paths in the catchment is facilitated by using a backward tracking formulation as the particles all start from their exit point, here the stream in the valley, and then move backward toward the boundaries of the domain. Assuming continuous and uniform infiltration in the system, the transit-time calculations with the backward approach show that almost all the subareas of the catchment supply the stream with water. The persistence of a few subareas not reached by the particles can be attributed to a backward formulation that is highly sensitive to the spatial resolution of the velocity fields close to the initial locations of the particles. Although the calculation mesh is refined close to the stream, the refinement is not sufficient for the backtracked particles to sample the whole catchment. This problem is intrinsic to the backward tracking approach. Transit times inferred by the backward tracking are usually short (less than 300 days) for a large portion of the catchment. Figure 8 shows that transit times longer than 500 days are associated only with water parcels that would have entered the subsurface compartment close to the crests of the catchment. Water falling in these elevated areas always infiltrates and should then run down along the steep slopes of the catchment to join the low-topography regions. The point is that the crests are often very dry, with low effective hydraulic conductivities and therefore low velocities explaining the long transit times estimated in these portions of the catchment.

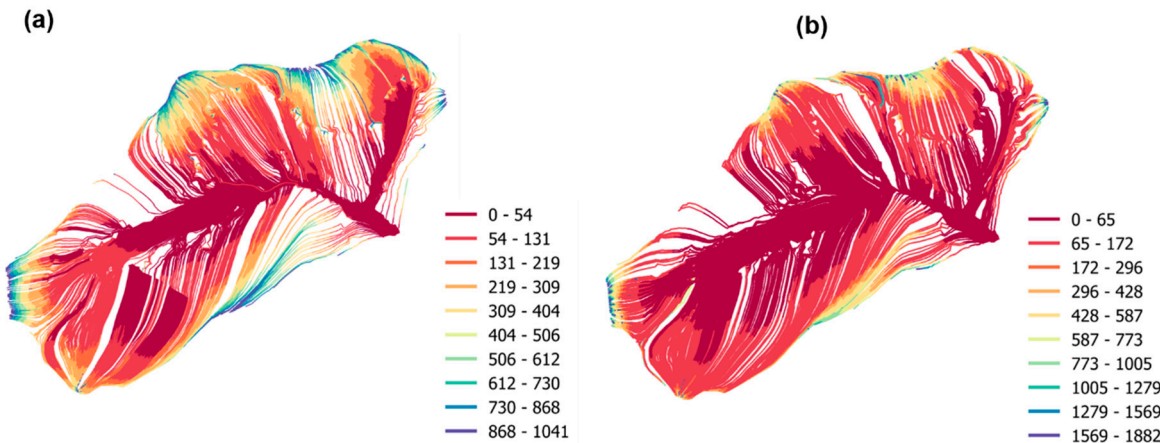

**Figure 8.** Maps of the flow paths inferred from backward particle tracking for the high-storage date HS9 (**a**) and low-storage date LS6 (**b**). The ten classes of times (days) along the flow paths are defined using equal counts in each class, thus setting the boundaries for classes that differ between HS9 and LS6.

The flow paths crossing the system for high- versus low-storage conditions are not very different. The transit times show more difference, low storage conditions being prone to large surface areas of the catchment associated with short transit times (Figure 8). The evolution of water storage within the catchment (see Figure 7) shows that the storage conditions are rather high prior to a low-storage date, this low storage being rapidly reached through drainage. In these conditions, particles that were launched during a low-storage period and then backtracked rapidly encountered high velocities (from high-storage periods) that spread them over large areas within short transit times. Conversely, the system passes from low-storage conditions to high-storage conditions through rapid infiltration and recharge. When launching backtracked particles from the stream under high-storage conditions, the particles rapidly encounter slower velocity fields that let them stay close to the stream for short transit times. Figure 8 also shows that the backward formulation gives artificially more weight to the long transit times because of water parcels that stay stuck in the dry-crest regions of the watershed with small velocity fields. This is especially visible for the HS9 calculation (see the map on the (a) panel of Figure 8) in which a non-negligible portion of the watershed is at the origin of transit times on the order of 800–1000 days.

This characteristic is confirmed in Figure 9, which compares transit-time distributions evaluated with both backward and forward formulations. The forward approach produces a highly skewed distribution, with a vast majority of short times and a flat tail of long times. The backward approach produces more uniform distributions with much higher proportions of long transit times. This difference is mainly due to the uniform weight given to each particle in the backward approach, in contrast with the particles being weighted by the rainfall intensity in the forward approach. In the backward approach, when a particle reaches an area with low velocity, its motion is limited but each motion counts evenly when it comes to computing the transit-time distribution. For its part, a forward approach injects diversely weighted particles all over the system that could never encounter (pass through areas of) slow velocities. In the backward tracking, one could consider that particles with high transit times have no statistical meaning, and therefore cut off the tail of the distribution beyond a given threshold. Figure 9 shows that changing the threshold from 500 to 200 days slightly shifts the backward distribution toward the forward distribution, but not enough to create convergence.

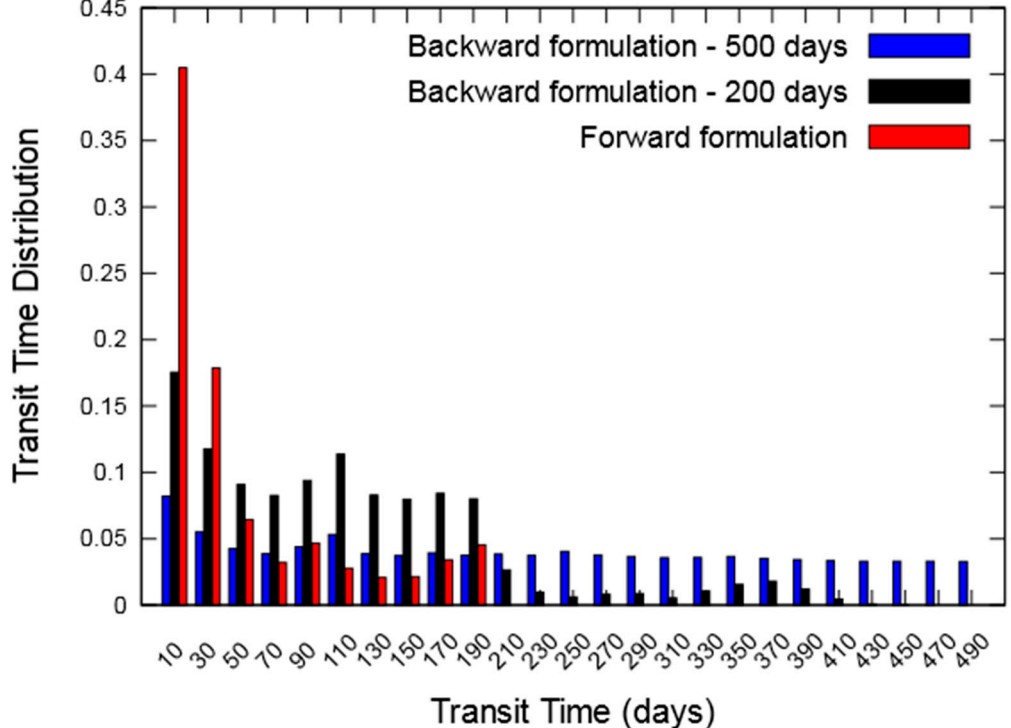

**Figure 9.** Comparison between transit-time distributions computed using backward and forward particle tracking. With the backward formulation, the distribution must be cut beyond a prescribed time to avoid excessive sampling of very long times associated with the stagnation of particles at the boundaries of the domain. The blue plot corresponds to a 500-day cut-off and the black plot to a 200-day cut-off.

The cumulative TTDs computed via forward tracking and applied to low- and high-storage conditions are presented in Figure 10. The blue curve represents the average of the cumulative distributions estimated for each of the low- (or high-) storage dates selected in Figure 7. The gray envelope illustrates the variability of these distributions by indicating the min and max probability of non-exceedance of a given transit time. The forward calculations account for the variability of both spatial and temporal distributions of water entering the system. The results suggested by the maps in Figure 8 are confirmed, with a vast majority of short transit times. Also, as suggested previously, when analyzing the flow-path patterns over the whole catchment, there is no clear difference between cumulative transit-time distributions at low- versus high-storage dates. The cumulative distributions have the same mean behavior with only slight differences regarding the min-max envelopes. The cumulative TTDs for low-storage conditions show a higher variability compared with those for high storage, probably due to the evolution of the storage before the date of reference. Low-storage conditions are often preceded by more varying hydrologic conditions than high-storage conditions (see Figure 7). Distributions in Figure 10 confirm that the overall system is stationary over time, always controlled by gravity-driven flow along steep slopes generating mostly short transit times, and with dry sub-areas close to the crests generating a few long transit times. Residence-time distributions (RTDs) reported in Figure 11 confirm this overall behavior, with shapes very similar to those of the cumulative TTDs.

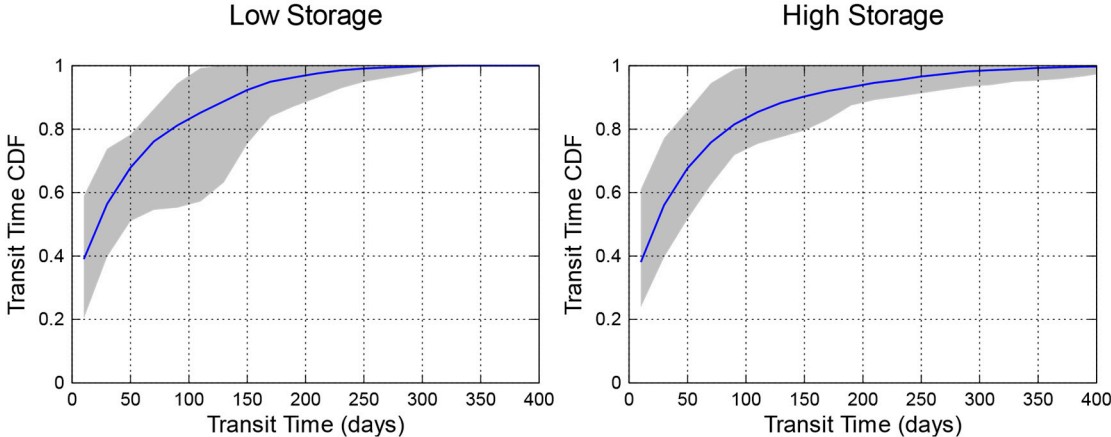

**Figure 10.** Cumulative transit-time distributions obtained via forward particle tracking for low-storage (LS) and high-storage (HS) conditions. The blue line represents the average distribution estimated over all the LS or HS dates considered (see Figure 7). The gray envelope shows the variability of the distributions by indicating the minimum and maximum cumulative probabilities of a given time.

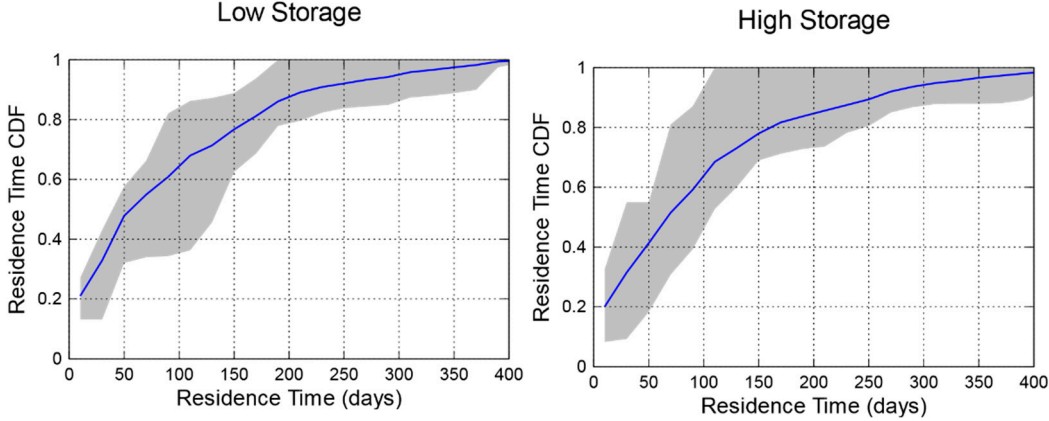

**Figure 11.** Cumulative residence-time distributions obtained via forward particle tracking for low-storage (LS) and high-storage (HS) conditions. The blue line represents the average distribution estimated over all the LS or HS dates considered (see Figure 7). The gray envelope shows the variability of the distributions by indicating the minimum and maximum cumulative probabilities of a given time.

Figure 12 shows the evolution of the so-called StorAge Selection (SAS) functions for low- and high-storage conditions. Several ways of representing the SAS functions can be used. Here, we plot for diverse successive prescribed times, considering the transit-time probability of non-exceedance of these prescribed times versus the equivalent residence-time probabilities. If both TTD and RTD were similar, this plot should render a straight line with a slope of one (where the non-exceedance probabilities of a prescribed time are similar), meaning that the catchment produces discharge distributed evenly within old and young water pools. If the plot is in the area where residence-time (non-exceedance) probabilities are larger than transit-time probabilities (the right side of "slope one" in the plots in Figure 12), the probability for a water parcel of a given age to remain stored within the catchment is higher than the probability for it to be released, meaning that the catchment preferentially samples old water to produce discharge. Conversely, if residence-time probabilities are smaller than transit-time probabilities (the left side of the plots in Figure 12), the probability for a water parcel of a given age to remain stored within the catchment is smaller than the probability for it to be released, meaning that the catchment preferentially samples young water to produce discharge. In Figure 12, all the SAS functions for both high- and low-storage conditions are under the one-to-one line, meaning that the catchment would always prefer to release old waters. This observation accords with the fact that the

catchment is small and that water parcels directly flow toward the valley without mixing with different water bodies (see the non-tortuous flow paths from crests to valley in Figure 8). This feature means that the water leaving the system at a given time is old water, having been stored in the contributive saturated area close to the stream; this water is pushed away via a piston-like effect by younger waters coming from upslope. This behavior can be explained by the very strong influence of the climate and the topography, with a catchment that is always wet on its slopes and that has a very steep topography. All this contributes to continuous and quite uniform injections of water being pushed toward the exit-point of the system by rapid and sweeping gravity-driven flow. To our knowledge, this kind of behavior is not common and has not been observed or modeled in previous studies that mostly demonstrate RTDs and TTDs being strongly controlled by storage.

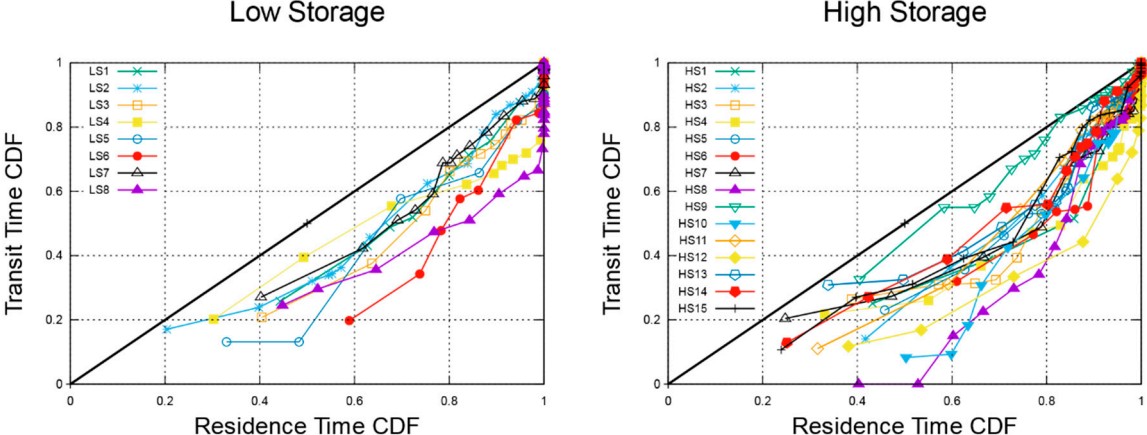

**Figure 12.** StorAge Selection (SAS) functions for low-storage (LS) and high-storage (HS) conditions. SAS functions are represented in a cumulative (CFD, cumulative density function) form by plotting for various times the cumulative transit-time probability versus the cumulative residence-time probability.

## 4. Conclusions

Transit-time and residence-time distributions are key descriptors of how a catchment stores and releases water that can bring insights into the hydrological functioning of mountainous catchments. In this study, a low-dimensional integrated hydrological model—that reduces the dimensionality of flow and thus computational costs—has been applied in combination with particle-tracking algorithms to assess the temporal variability of TTDs and RTDs on the Strengbach catchment, a mountainous catchment located in the Vosges mountains, in France, and a node of the French National network of Critical-Zone Observatories. The model is calibrated using an innovative approach that combines streamflow and magnetic resonance sounding measurements. MRS provides data about the water content of the subsurface, making it possible to condition a hydrological model via "non-intrusive" monitoring and without using classical head data about the subsurface. This procedure was proven to be very efficient, because it ensures that the model is able both to reproduce the overall response of the catchment at the outlet and also capture the local dynamics within the catchment. The transient velocity fields are used as an input for particle-tracking algorithms allowing us to estimate TTDs and RTDs at a given time.

The response of Strengbach catchment regarding TTDs and RTDs is stationary over time and independent of the amount of water stored in the catchment, with a system that prefers to use old water to produce runoff. The stationary behavior, a characteristic not commonly discussed in previously published studies, is related to the small size of the catchment and to specific climatic and topographic conditions prevailing at the Strengbach catchment. Rainfall temporal distribution makes the catchment rather wet all year long. Infiltrated water is quickly driven close to the stream through gravity effects along the steep slopes and then pushes rather old water out of the system. The comparison between forward and backward particle-tracking algorithms has shown that forward approaches should be

used to estimate the temporal variability of RTDs or TTDs, since they allow particles to carry a weight related to the rainfall rate as well as the location where they enter the system. Backward approaches are interesting insofar as they can be used to track water parcels defining an exact exit time; but all particles have the same weight in the computation of the TTDs, with this feature artificially shifting the distribution toward long transit times.

Finally, this study exemplifies that physically based modeling approaches can be a relevant alternative to tracer-based approaches, which are the ones most often used in the literature, to study the temporal variation of TTDs and RTDs. Using distributed models allows the evolution of TTDs and RTDs to be interpreted in a mechanistic manner. This can complement the approaches that use high-frequency measurements of tracers in precipitation and stream water. To our knowledge, the use of an integrated model in combination with high-frequency tracer information on the same catchment has not yet been performed. This could be an interesting avenue for future research, since these two approaches should converge on the same conclusions.

**Author Contributions:** All the authors have contributed to the development of the NIHM model, its application to the Strengbach catchment. All authors have read and agree to the published version of the manuscript. Conceptualization, mainly S.W. and F.D., methodology, S.W., N.L., F.D.; software, S.W., B.J.; validation, S.W., B.J., N.L.; writing—original draft preparation, S.W., F.D.; writing—review and editing, F.D.; funding acquisition, S.W., F.D., N.L.

**Funding:** The authors would like to thank the French Ministry of Agriculture for funding Benjamin Jeannot's Ph.D.

**Acknowledgments:** The authors would like to acknowledge the OZCAR network for the scientific animation that has led to this study and the OHGE observatory for providing physical space for the study as well as access to the data.

**Conflicts of Interest:** The authors declare no conflict of interest.

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
