# Peer review of "Variability of Water Transit Time Distributions at the Strengbach Catchment (Vosges Mountains, France) Inferred Through Integrated Hydrological Modeling and Particle Tracking Algorithms"

_water, doi:10.3390/w11122637_

Round 1
Reviewer 1 Report
I reviewed this article “Variability of water transit time distributions at the Strengbach catchment (Vosges Mountains, France) inferred through integrated hydrological modelling and particle tracking algorithms” which is an excellent piece of work which significantly contribute into knowledge. Unfortunately, the article is not well presented, and some improvement needed in the modelling results where authors showed the results i.e. Figs 5 & 6.
I would like to review this article again after following suggested changes:
The authors should write about the model with its implications in the introduction part of the paper, the remaining part belongs to the methodology. The overall introduction section of the paper is poorly written. The data part should be removed from the study site section. The authors must add the model efficiency indices section after the model description i.e. Nash-Sutcliffe etc. The observed flow shows no flow for a certain period in 1996 (bottom figure 6) and the simulated flow is not corresponding to the observed flow. Zero flow is possible in a small catchment with a minimum baseflow contribution, I would suggest the authors look again the model used parameters and the soil physical properties data. Also, the authors need to show the rainfall data along the y-axis to see model sensitivity to the rainfall events. If possible, in the model, plot flow against the soil moisture for a particular day. Authors should replace RMSE with other model efficiency indices like a log of Nash-Sutcliffe, this is needed for the low flows where the modelled streamflow is overestimating in comparison to the baseline. To find the robustness of the model parameters, I would suggest carrying the model uncertainty analysis. The results and conclusions section of the article needs improvementI will like to review the article after the suggested changes.
Author Response
We thank for their time and efforts the Associate Editor and the two reviewers who perused the first version of our manuscript. Before proceeding with specific responses to the reviewers’ comments, we would raise a general statement on how we improved the revised version of the manuscript. The first version was rated as needing major revisions by Rev #1 mainly on the features that it was poorly written (we understand in view of the mark on the rating form, for “poor usage of English”), and that the model was inappropriate to simulate null flow rates in streams…
First, the revised manuscript was re-edited by a professional editing service (WordRu, invoice and certificate available on request). This service only brought very few modifications, simply changing a word for another, or slightly changing the structure of a sentence to better emphasize an idea. We clearly doubt that these modifications were absolutely necessary to raise the paper to something acceptable for publication. That being said, the revised manuscript should now follow the strict standards of English language.
Second, a rapid look at the original figures in the manuscript clearly shows that the inability of the model to reproduce part of data over short periods, is only associated with lack of data (due to failures in the monitoring systems), and values set to zero for these gaps. In any case, it is not because a model would fail to reproduce 5% of data that it can be labelled as inappropriate, then needing for coming back to its parameterization, etc. By definition, a model is not reality, we only expect that it can reproduce pieces of this reality (provided that the observations of reality are not flawed).
At this stage of the review process, we are inclined to think that the Associate Editor could first account for the comments above, then read how we answer to the reviewers’ comments and finally state whether or not the revised version needs for another round of review.
The following details our answers to reviewers’ comments. Theses answers are presented in italic, to distinguish them from the reviewers’ comments
Reviewer 1:
I reviewed this article “Variability of water transit time distributions at the Strengbach catchment (Vosges Mountains, France) inferred through integrated hydrological modelling and particle tracking algorithms” which is an excellent piece of work which significantly contribute into knowledge.
Unfortunately, the article is not well presented, and some improvement needed in the modelling results where authors showed the results i.e. Figs 5 & 6.
We let the reviewer judging how harsh can be his rating on the paper as “needing major revisions”, when the rest of the review mainly mentions that: 1- the model should appear in the introduction and not in the methodology, and 2- two figs, on the dozen that counts the paper, need for two cosmetics. Those are adding a rainfall history, and removing slight portions of a plot because the reviewer did not grasp that an abrupt (instantaneous) decrease of flow rate values down to exactly zero (an awkward behavior for a hydrological variable) translated in the plots the lack of data over short periods.
The authors should write about the model with its implications in the introduction part of the paper, the remaining part belongs to the methodology.
For a contribution in the domain of applications and dealing with field data, the model, its settings, and its parameterization obviously correspond to methodology. A few words on various models could appear in an Introductive Section, if the topic of the study were, for example, to compare a new development with previous ones. In the present case, it is reminded that the main topic is about evaluating transit times via a modeling approach, a technique which is barely employed, except for synthetic test cases. As a technique for identifying various hydrological behaviors, the model belongs to the methodology.
The overall introduction section of the paper is poorly written.
As told above, it is not the opinion of the professional editing service that perused the original manuscript to help us building the revised version. If here “poor” means that the ideas concealed in the introduction are obsolete, that they are not well presented, that there exist lacks in the state of the art on the topic, etc., Rev #1 would have really helped us in being more specific and giving us a few suggestions.
The data part should be removed from the study site section.
It is of common practice in a vast majority of the papers in the literature to regroup the presentation of data and of the study site in the same Section (e.g.; Kaandrop et al, WRR, 2018; Benettin et al, WRR, 2017; Benettin et al, WRR, 2015, …). We chose to do the same because Section 2 (dedicated to the material and methods) already included 5 sub-sections. We believe that this choice improves the readability of the paper, and in a paper for a special issue on observatories of the Critical Zone, observatories go with sites and observations with data.
The authors must add the model efficiency indices section after the model description i.e. Nash-Sutcliffe tc.
We added a short paragraph to rapidly depict the indicators employed, even though they are now well-known in the ongoing literature about Hydrology and can be understood by any reader without further details on their calculation.
The observed flow shows no flow for a certain period in 1996 (bottom figure 6) and the simulated flow is not corresponding to the observed flow. Zero flow is possible in a small catchment with a minimum baseflow contribution, I would suggest the authors look again the model used parameters and the soil physical properties data.
The period in 1996 mentioned by Rev #1 is not a period of no flow in the stream. The abrupt break down of flow rates as step functions in the original plots was easily foreseeable as lack of data (due to failure of the automatic monitoring device). It is right that we did not mention these lacks of data in the original manuscript. This is now clearly specified (Lines 442-443) in the revised version of the paper. That being said, all the comments of Rev #1 regarding the inability of the model to mimic null flow rates in the stream, the fair scientific results associated with this inability, and the need for re-handling our model parameterization, fall down. We apologize for having probably puzzled Rev #1; the revised version both mentions the lack of data, and the artificial step-shaped curves of flow rates down to zero in the plots have been removed.
Also, the authors need to show the rainfall data along the y-axis to see model sensitivity to the rainfall events.
Rainfall data are now reported in Fig. 5 to exemplify the model sensitivity to rainfall. The plot of rainfall versus time was done only on this specific figure to avoid overloading the other ones.
If possible, in the model, plot flow against the soil moisture for a particular day.
We do not believe that this kind of plot would bring insights on the main topic of inferring transit and residence time distributions over a small watershed. Transit and residence times are the consequence of water parcel trajectories through the various compartments of the watershed and over many successive days. The soil moisture of one day in the vadose zone mimicked by the model is of few interest, except for the model itself, for example, to check on its ability to reproduce local data. Unfortunately, local data on soil moisture are not available in the catchment. We decided to let the revised manuscript with a reasonable number of figures and not to mislead any further reader with marginal comments.
Authors should replace RMSE with other model efficiency indices like a log of Nash-Sutcliffe, this is needed for the low flows where the modelled streamflow is overestimating in comparison to the baseline.
Nash-Sutcliffe coefficients have been added in all the figures where simulated and observed discharges are compared.
To find the robustness of the model parameters, I would suggest carrying the model uncertainty analysis.
The manuscript is submitted for tentative publication in a special issue devoted to observatories as tools to understand mechanisms and processes occurring in the so-called Critical Zone. This aim is obviously incentive to applied studies that show, for example, how more theoretical approaches and models can cope with concrete-case studies. A thorough sensitivity analysis (and not simply collecting a few results to roughly bound the variation ranges of parameters) would need model mathematical decomposition (as Chaos Polynomial Expansion, for example) and/or automatic inversion procedures. Both are no yet available for NIHM (our model), even though we are in the process of building an inversion procedure dedicated to NIHM. In addition, one can mention that for any application of a model to an actual system, performing a sensitivity analysis is only meaningful for systems with a large amount of conditioning data (which we do not have here). Otherwise, any attempt to check on model parameter distributions cannot be interpreted either in terms of model robustness (and repeatability) or as lack of data to witness how the system behaves. We hope that further investigations with MRS data repeated over time and space will help us to conduct a sensitivity analysis, a modeling exercise which never appeared in the literature for integrated hydrological models.
The results and conclusions section of the article needs improvement.
This comment is evasive and hardly understandable as such. That being said, the other reviewer (Rev #2) made a few suggestions regarding results and conclusions, especially in asking us to shorten the writing and emphasize key findings. We followed this suggestion in the revised manuscript, and by the way, we hope that the modifications will satisfactorily answer to Rev #1 requests.
Reviewer 2 Report
This article is focused on studying the variability of water transit time distributions at the Strengbach catchment. The article is well-written and has an interesting topic. Some minor revisions are needed to make the work ready for publication.
1- The study sire is very small. Is there any potential impact on the results and findings. I think author needs to elaborate on the potential impacts of catchment size on their study.
2- Lines 188 to 191. It is needed to give equations a numbering system, so they can be referred to.
3- Figure 4: I suggest to change the markers on the figure (especially the star-shape one) as they are so big.
4- Figure 6: first of all it is better to use letters a,b,c to address the graphs under this figure rather than using up,left,right, ...
Secondly, in the down-left one, simulated time span is different with observed one. Is it a mistake in generating the figure? revision is needed.
5- Figure 6: besides RMSE, a very good parameter that can nicely gauge the goodness-of-fit is the Nash-Sutcliffe coefficient of Efficiency as it can give the readers an understanding of the performance better than RMSE. This parameter is also a better assessor for the extreme values.
6- Figure 7 and 8: same comment about labeling the figures. Please use letters rather than left/right.
7- The conclusion is lengthy. It is recommended to shorten it with the emphasis on the key findings of the work.
Author Response
We thank for their time and efforts the Associate Editor and the two reviewers who perused the first version of our manuscript. Before proceeding with specific responses to the reviewers’ comments, we would raise a general statement on how we improved the revised version of the manuscript. The first version was rated as needing major revisions by Rev #1 mainly on the features that it was poorly written (we understand in view of the mark on the rating form, for “poor usage of English”), and that the model was inappropriate to simulate null flow rates in streams…
First, the revised manuscript was re-edited by a professional editing service (WordRu, invoice and certificate available on request). This service only brought very few modifications, simply changing a word for another, or slightly changing the structure of a sentence to better emphasize an idea. We clearly doubt that these modifications were absolutely necessary to raise the paper to something acceptable for publication. That being said, the revised manuscript should now follow the strict standards of English language.
Second, a rapid look at the original figures in the manuscript clearly shows that the inability of the model to reproduce part of data over short periods, is only associated with lack of data (due to failures in the monitoring systems), and values set to zero for these gaps. In any case, it is not because a model would fail to reproduce 5% of data that it can be labelled as inappropriate, then needing for coming back to its parameterization, etc. By definition, a model is not reality, we only expect that it can reproduce pieces of this reality (provided that the observations of reality are not flawed).
At this stage of the review process, we are inclined to think that the Associate Editor could first account for the comments above, then read how we answer to the reviewers’ comments and finally state whether or not the revised version needs for another round of review.
The following details our answers to reviewers’ comments. Theses answers are presented in italic, to distinguish them from the reviewers’ comments.
Reviewer 2:
This article is focused on studying the variability of water transit time distributions at the Strengbach catchment. The article is well-written and has an interesting topic. Some minor revisions are needed to make the work ready for publication.
1- The study site is very small. Is there any potential impact on the results and findings? I think author needs to elaborate on the potential impacts of catchment size on their study.
The fact that the size of the catchment impacts our findings is obvious and was already mentioned in the original version of the manuscript. As size, topography and climatic characteristics are intrinsically intertwined, it is hard to elaborate on the impact of the size alone. The literature reports on studies carried out over catchment with sizes similar to the Strengbach but that show very different hydrologic behavior as the consequence of different geologic or climatic characteristics. It is obvious that the small size of the Strengbach catchment, its steep slopes, and a very shallow permeable aquifer are features that intuitively go with short transit times. We mentioned that already, but pushing beyond the reasoning as a generalization to small catchments would be conjectured.
2- Lines 188 to 191. It is needed to give equations a numbering system, so they can be referred to.
Changed in the revised version.
3- Figure 4: I suggest to change the markers on the figure (especially the star-shape one) as they are so big.
The size of the yellow stars in Fig. 4 has been reduced as suggested.
4- Figure 6: first of all it is better to use letters a,b,c to address the graphs under this figure rather than using up,left,right, ...
Changed in the revised version.
Secondly, in the down-left one, simulated time span is different with observed one. Is it a mistake in generating the figure? revision is needed.
The figure was modified according to Rev #2 comment.
5- Figure 6: besides RMSE, a very good parameter that can nicely gauge the goodness-of-fit is the Nash-Sutcliffe coefficient of Efficiency as it can give the readers an understanding of the performance better than RMSE. This parameter is also a better assessor for the extreme values.
Nash-Sutcliffe coefficients have been added in all the figures where simulated and observed discharges are compared.
6- Figure 7 and 8: same comment about labeling the figures. Please use letters rather than left/right.
Changed in the revised version.
7- The conclusion is lengthy. It is recommended to shorten it with the emphasis on the key findings of the work.
The conclusion was shortened as recommended.